# Adaptive Market Making via Online Learning

**Jacob Abernethy***
Computer Science and Engineering
University of Michigan
jabernet@umich.edu

**Satyen Kale**
IBM T. J. Watson Research Center
sckale@us.ibm.com

## Abstract

We consider the design of strategies for *market making* in an exchange. A market maker generally seeks to profit from the difference between the buy and sell price of an asset, yet the market maker also takes exposure risk in the event of large price movements. Profit guarantees for market making strategies have typically required certain stochastic assumptions on the price fluctuations of the asset in question; for example, assuming a model in which the price process is mean reverting. We propose a class of "spread-based" market making strategies whose performance can be controlled even under worst-case (adversarial) settings. We prove structural properties of these strategies which allows us to design a master algorithm which obtains low regret relative to the best such strategy in hindsight. We run a set of experiments showing favorable performance on recent real-world stock price data.

## 1 Introduction

When a trader enters a market, say a stock or commodity market, with the desire to buy or sell a certain quantity of an asset, how is this trader guaranteed to find a counterparty to agree to transact at a reasonable price? This is not a problem in a liquid market, with a deep pool of traders ready to buy or sell at any time, but in a thin market the lack of counterparties can be troublesome. A rushed trader may even be willing to transact at a worse price in exchange for immediate execution.

This is where a *market maker* (MM) can be quite useful. A MM is any agent that participates in a market by offering to both buy *and* sell the underlying asset at any time. To put it simply, a MM consistently guarantees liquidity to the marketplace by promising to be a counterparty to any trader. The act of market making has both potential benefits and risks. For one, the MM bears the risk of transacting with better-informed traders that may know much more about the movement of the asset's price, and in such scenarios the MM can take on a large inventory of shares that it may have to offload at a worse price. On the positive side, the MM can profit as a result of the *bid-ask spread*, the difference between the MM's buy price and sell price. In other words, if the MM buys 100 shares of a stock from one trader at a price of $p$, and then immediately sells 100 shares of stock to another trader at a price of $p + \Delta$, the MM records a profit of $100\Delta$.

This describes the central goal of a profitable market making strategy: minimize the inventory risk of large movements in the price while simultaneously aiming to benefit from the bid-ask spread. The MM strategy has a *state*, which is the current inventory or holdings, receives as input order and price data, and must decide what quantities and at what prices to offer in the market. In the present paper we assume that the MM interacts with a *continuous double auction* via an *order book*, and the MM can place both *market and limit orders* to the order book.

A number of MM strategies have been proposed, and in many cases certain profit/loss guarantees have been given. But to the best of our knowledge all such guarantees (aside from [4]) have required

*stochastic assumptions* on the traders or the sequence of price fluctuations. Often, e.g., one needs to assume that the underlying price process exhibits a *mean reverting* behavior to guarantee profit.

In this paper we focus on constructing MM strategies that achieve *non-stochastic* guarantees on profit and loss. We begin by proposing a class of market making strategies, parameterized by the choice of bid-ask spread and liquidity, and we establish a data-dependent expression for the profit and loss of each strategy at the end of a sequence of price fluctuations. The model we consider, as well as the aforementioned class of strategies, builds off of the work of Chakraborty and Kearns [4]. In particular, we assume the MM is given an exogenously-specified price time series that is revealed online. We also assume that the MM is able to make and cancel orders after every price fluctuation.

We extend the prior work [4] by considering the problem of *online learning* among this parameterized class of strategies. Performance is measured in terms of *regret*, which is the difference between the total value of the learner's algorithm and that of the best strategy in hindsight. While this problem is related to the problem of learning from expert advice, standard algorithms assume that the experts have *no state*; i.e. in each round, the cost of following any given expert's advice is the same as the cost to that expert. This is not the case for online learning of the bid-ask spread, where the state, represented by the inventory of each strategy, affects the payoffs. We can prove however that due to the combinatorial structure of these strategies, one can afford to switch state with bounded cost. Using these structural properties we prove the following main result of this paper:

**Theorem 1** *There is an online learning algorithm, that, under a bounded price volatility assumption (see Defintion 1) has $O(\sqrt{T})$ regret after $T$ trading periods to the best spread-based strategy.*

Experimental simulations of our online learning algorithms with real-world price data suggest that this approach is quite promising; our algorithm frequently performs nearly as well as the best strategy, and is often superior. Such empirical results provides some evidence that regret minimization techniques are well-suited for adaptively setting the bid-ask spread.

**Related Work**    Perhaps the most popular model to study market making has been the Glosten-Milgrom model [11]. In this setting the market is facilitated by a *specialist*, a monopolistic market maker that acts as the middle man for all trades. There has been some work in the Computer Science literature that has considered the sequential decision problem of the specialist [8, 10], and this work was extended to look at the more modern order book mechanism [9]. In our model traders interact directly with an order book, not via a specialist, and the prices are set exogenously as in [4].

Over the past ten years that has been a burst of research within the AI and EconCS community on the design of *prediction markets* in which traders can bet on the likelihood of future uncertain events (like horse races, or elections). Much of this started with a couple of key results of Robin Hanson [12, 13] who described how to design subsidized prediction markets via the use of *proper scoring rules*. The key technique was a method to design an automated market maker, and there has been much work on facilitating this using mechanisms based on shares (i.e. Arrow-Debreu securities). There is a medium-sized literature on this topic by now [6, 5, 1, 2] and we mention only a selection. The key difference between the present paper and the work on designing prediction markets is that our techniques are solely focused on profit and risk, and not on other issues like price discovery or information aggregation. Recent work by Della Penna and Reid [19] considered market making as a the multi-armed bandit problem, and this is a notable exception where profit was the focus.

This "non-stochastic" approach we take to the market making problem echos many of the ideas of Cover's results on Universal Portfolio algorithms [20], an area that has received much followup work [16, 15, 14, 3, 7] given its robustness to adversarially-chosen price fluctuations. But these algorithms are of the "market taking" variety, that is they actively rebalance their portfolio on a daily basis. Moreover, the goal of the Universal Portfolio is to get low regret with respect to the best fixed mixture of investments, rather than the best bid-ask spread which is aim of the present work.

## 2    The Market Execution Framework

We now present our market model formally. We will consider the buying and selling of a single security, say the stock of Microsoft, over the course of some time interval. We assume that all events in the market take place at discrete points in time throughout this day. At each time period $t$ a

*market price* $p_t$ is announced to the world. In a typical stock exchange this price will be rounded to a discrete value; historically stock prices were quoted in $\frac{1}{8}$'s of a dollar, although now they are quoted in pennies. We let $\delta$ be the *discretization parameter* of the exchange, and for simplicity assume $\delta = 1/m$ for some positive integer $m$. Now let $\Pi$ be the set of discrete prices within some feasible range, $\Pi := \{\delta, 2\delta, 3\delta, \ldots, (\frac{M}{\delta} - 1)\delta, M\}$, where $M$ is some reasonable bound on the largest price.

A trading strategy maintains two state variables at the beginning of every time period $t$: (a) the *holdings* or *inventory* $H_t \in \mathbb{R}$, representing the amount of stock that the strategy is long or short at the beginning of time period $t$ ($H_t$ will be negative if the strategy is short); (b) the *cash* $C_t \in \mathbb{R}$ of the strategy, representing the money earned or lost by the investor at that time. Initially we have $C_1 = H_1 = 0$. Note that when $C_t < 0$ this is not necessarily bad, it simply means the investor has borrowed money to purchase holdings, often referred to as "trading on margin".

Let us now consider the trading mechanism at time $t$. For simplicity we assume there are two types of trades that can be executed, and each will change the cash and holdings at the following time period. By default, set $H_{t+1} \leftarrow H_t$ and $C_{t+1} \leftarrow C_t$. Then the trading strategy can execute any subset of the following two actions:

- **Market Order**: At time $t$ the posted price is $p_t$ and the trader executes a trade of $X$ shares, with $X \in \mathbb{R}$. In this case we update the cash as $C_{t+1} \leftarrow C_{t+1} - p_t X$ and $H_{t+1} \leftarrow H_{t+1} + X$. Note that if $X < 0$ then this is a short sale in which case the trader's cash increases[1]

- **Limit Order**: Before time period $t$, the trader submits a demand schedule $L_t : \Pi \to \mathbb{R}_+$, where it is assumed that $L_t(p_{t-1}) = 0$. For every price $p \in \Pi$ with $p < p_{t-1}$, the value $L_t(p)$ is the number of shares the trader would like to *buy* at a price of $p$. For every $p > p_{t-1}$ the value $L_t(p)$ is the number of shares the trader would like to *sell* at a price of $p$. One should interpret a limit order in terms of "posting shares to the order book": these shares are up for sale (and/or purchase) but the order will only be executed if the price moves.

  In round $t$ the posted price becomes $p_t$ and it is assumed that all shares offered at any price between $p_{t-1}$ and $p_t$ are transacted. More specifically, we have two cases:

  - If $p_t > p_{t-1}$ then for each $p \in \Pi$ with $p_{t-1} < p \leq p_t$ we update $C_{t+1} \leftarrow C_{t+1} + pL_t(p)$ and $H_{t+1} \leftarrow H_{t+1} - L_t(p)$;
  - Else if $p_t < p_{t-1}$ then for each $p \in \Pi$ with $p_t \leq p < p_{t-1}$ we update $C_{t+1} \leftarrow C_{t+1} - pL_t(p)$ and $H_{t+1} \leftarrow H_{t+1} + L_t(p)$.

It is worth noting market orders are quite different from limit orders. A limit order is a passive action in the market, the trader simply states that he would be willing to trade a number of shares at a range of different prices. But if the market does not move then no transactions occur. The market order is a much more direct action to take, the transaction is guaranteed to execute at the current market price. The market order has the downside that the trader does not get to specify the price at which he would like to trade, contrary to the limit order. Roughly speaking, an MM strategy will generally interact with the market via limit orders, since the MM is simply hoping to profit from liquidity provision. But the MM may at times have to place market orders to balance inventory to control risk.

We include one more piece of notation, the value of the strategy's portfolio $V_{t+1}$ at the end of time period $t$, which can be defined explicitly in terms of the cash, holdings, and current market price: $V_{t+1} := C_{t+1} + p_t H_{t+1}$. In other words, $V_{t+1}$ is the amount of cash the strategy would have if it liquidated all holdings at the current market price.

**Assumptions of our model.** In the described framework we make several simplifying assumptions on the trading execution mechanism, which we note here.

**(1)** The trader pays neither transaction fees nor borrowing costs when his cash balance is negative.
**(2)** Market orders are executed at exactly the posted market price, without "slippage" of any kind. This suggests that the market is very liquid relative to the actions of the MM.
**(3)** The market allows the buying and selling of fractional shares.

**(4)** The price sequence is "exogenously" determined, meaning that the trades we make do not affect the current and future prices. This assumption has been made in previous results [4] and it is perhaps quite strong, especially if the MM is providing the bulk of the liquidity. We leave it for future work to consider the setting with a non-exogenous price process.

**(5)** Unexecuted limited orders are cancelled before the next period. That is, for any $p$ not lying between $p_{t-1}$ to $p_t$ it is assumed that the $L_t(p)$ untransacted shares at price $p$ are removed from the order book. This is just notational convenience: the MM can resubmit these shares via $L_{t+1}$.

## 3 Spread-based Strategies

In this section we present a class of simple market making strategies which we refer to as *spread-based strategies* since they maintain a fixed bid-ask spread throughout. We then prove some structural properties on this class of strategies. We only give proof sketches for lack of space; all proofs can be found in an appendix in the supplementary material.

### 3.1 Spread-based strategies.

We consider market making strategies parameterized by a window size $b \in \{\delta, 2\delta, \ldots, B\}$, where $B$ is a multiple of $\delta$. Before round $t$, the strategy $S(b)$ selects a window of size $b$, viz. $[a_t, a_t + b]$, starting with $a_1 = p_1$. For some fixed liquidity density parameter $\alpha$, it submits a buy order of $\alpha$ shares at every price $p \in \Pi$ such that $p < a_t$ and a sell order $\alpha$ shares at every price $p \in \Pi$ such that $p > a_t + b$. Depending on the price in the trading period $p_t$, the strategy adjusts the next window by the smallest amount necessary to include $p_t$.

---

**Algorithm 1** Spread-Based Strategy $S(b)$

---
1: Receive parameters $b > 0$, liquidity density $\alpha > 0$, inital price $p_1$ as input. Initialize $a_1 := p_1$.
2: **for** $t = 1, 2, \ldots, T$ **do**
3:     Observe market price $p_t$
4:     **If** $p_t < a_t$ **then** $a_{t+1} \leftarrow p_t$
5:     **Else If** $p_t > a_t + b$ **then** $a_{t+1} \leftarrow p_t - b$
6:     **Else** $a_{t+1} \leftarrow a_t$
7:     Submit limit order $L_{t+1}$: $L_{t+1}(p) = 0$ if $p \in [a_{t+1}, a_{t+1} + b]$, else $L_{t+1}(p) = \alpha$.
8: **end for**

---

The intuition behind a spread-based strategy is that the MM waits for the price to deviate in such a way that it leaves the window $[a_t, a_t + b]$. Let's say the price suddenly drops below $a_t$ and we get $p_t = a_t - k\delta$ for some positive integer $k$ such that $k\delta < b$. As soon as this happens some transactions occur and the MM now has holdings of $k\alpha$ shares. That is, the MM will have purchased $\alpha$ shares at each of the prices $a_t - \delta, a_t - 2\delta, \ldots, a_t - k\delta$. On the following round the MM updates his limit order $L_{t+1}$ to offer to sell $\alpha$ shares at each of the price levels $a_t + b - (k-1)\delta, a_t + b - (k-2)\delta, \ldots$. This gives a natural matching between shares that were bought and shares that are offered for sale, with the sale price being *exactly* $b$ higher than the purchased price. If, at a later time $t' > t$, the price rises so that $p_{t'} \geq a_t + b + \delta$ then all shares bought previously are sold at a profit of $kb\alpha$.

We now give a very useful lemma, that essentially shows that we can calculate the profit and loss of a spread-based strategy on two factors: (a) how much the spread window moves throughout the trading period, and (b) how far away the final price is from the initial price. A sketch of the proof is provided, but the complete version is in the Appendix.

**Lemma 1** *The value of the portfolio of $S(b)$ at time $T$ can be bounded as*

$$V_{T+1} \geq \frac{\alpha}{\delta} \left( \sum_{t=1}^{T} \frac{b}{2}|a_{t+1} - a_t| - (|a_{T+1} - a_1| + b)^2 \right)$$

PROOF:[Sketch] The proof of this lemma is quite similar to the proof of Theorem 2.1 in [4]. The main idea is given in the intuitive explanation above: we can match pairs of shares that are bought

and sold at prices that are $b$ apart, thus registering a profit of $b$ for each such pair. We can relate these matched pairs to the $a_t$'s, and the unmatched stock transactions to the difference $|a_{T+1} - a_1|$, yielding the stated bound. □

In other words, the risk taken by all strategies is roughly the same ($\frac{1}{2}|p_{T+1} - p_1|^2$ up to an additive constant in the quadratic term). But the revenue of the spread-based strategy scales with two quantities: the size of the window $b$ but also the total movement of the window. This raises an interesting tradeoff in setting the $b$ parameter, since we would like to make as much as possible on the movement of the window, but by increasing $b$ the window will get "pushed around" a lot less by the fluctuating price.

We now make some convenient normalization. Since for every unit price change, the strategies trade $\alpha/\delta$ shares, in the rest of the paper, without loss of generality, we may assume that $\alpha = 1$ and $\delta = 1$ (by appropriately changing the unit of currency). The regret bounds for general $\alpha$ and $\delta$ scale up by a factor of $\frac{\alpha}{\delta}$.

## 3.2 Structural properties of spread-based strategies.

It is useful to prove certain properties about the proposed spread-based strategies.

**Lemma 2** *Consider any two strategies $S(b)$ and $S(b')$ with $b' < b$. Let $[a'_t, a'_t + b']$ and $[a_t, a_t + b]$ denote the intervals chosen by $S(b)$ and $S(b')$ at time $t$ respectively. Then for all $t$, we have $[a'_t, a'_t + b'] \subset [a_t, a_t + b]$.*

PROOF:[Sketch] This is easy to prove by induction on $t$, via a simple case analysis on where $p_t$ lies in relation to the windows $[a'_t, a'_t + b']$ and $[a_t, a_t + b]$. □

**Lemma 3** *For any strategy $S(b)$, its inventory at time $t$, $H_t$, equals $a_1 - a_t$.*

PROOF:[Sketch] Again using case analysis on where $p_t$ lies in relation to the window $[a_t, a_t + b]$, we can show that $H_t + a_t$ is an invariant. Thus, $H_t + a_t = H_1 + a_1 = a_1$, and hence $H_t = a_1 - a_t$. □

The following corollary follows easily:

**Corollary 1** *For any round $t$, consider any two strategies $S(b)$ and $S(b')$ with $b' < b$, with inventories $H_t$ and $H'_t$ respectively. Then $|H_t - H'_t| \leq b - b'$.*

PROOF: By Lemma 3 we have $|H_t - H'_t| = |a_1 - a'_1 + a'_t - a_t| \leq b - b'$, since $[a'_1, a'_1 + b'] \subset [a_1, a_1 + b]$ and by Lemma 2 $[a'_t, a'_t + b'] \subset [a_t, a_t + b]$. □

**Definition 1 ($\Delta$-bounded volatility)** *A price sequence $p_1, p_2, \ldots, p_T$ is said to have $\Delta$-bounded volatility if for all $t \geq 2$, we have $|p_t - p_{t-1}| \leq \Delta$.*

We assume from now that the price sequence has $\Delta$-bounded volatility. Suppose now that we have a set $\mathcal{B}$ of $N$ window sizes $b$, all bounded by $B$. In the rest of the paper, all vectors are in $\mathbb{R}^N$ with coordinates indexed by $b \in \mathcal{B}$. For every $b \in \mathcal{B}$, at the end of time period $t$, let its inventory be $H_{t+1}(b)$, cash value be $C_{t+1}(b)$, and total value be $V_{t+1}(b)$. These quantities define the vectors $H_{t+1}$, $C_{t+1}$ and $V_{t+1}$. The following lemma shows that the change in the total value of different strategies in any round is similar.

**Lemma 4** *Define $G = 2\Delta B + \Delta^2$. In round $t$, $H = \min_{b \in \mathcal{B}}\{H_t(b)\}$. Then for any strategy $S(b)$, we have*

$$|(V_{t+1}(b) - V_t(b)) - (H(p_t - p_{t-1}))| \leq G.$$

*Thus, for any two window sizes $b$ and $b'$, we have*

$$|(V_{t+1}(b) - V_t(b)) - (V_{t+1}(b') - V_t(b'))| \leq 2G.$$

PROOF:[Sketch] Since $|p_t - p_{t-1}| \leq \Delta$, each strategy trades at most $\Delta$ shares, at prices between $p_{t-1}$ and $p_t$. Next, by Corollary 1, for any strategy $|H_t(b) - H| \leq B$. Using these bounds, and the definitions of the total value, some calculations give the stated bounds. □

## 4 A low regret meta-algorithm

Recall that we have a set $\mathcal{B}$ of $N$ window sizes $b$, all bounded by $B$. We want to design a low-regret algorithm that achieves almost as much payoff as that of the best strategy $S(b)$ for $b \in \mathcal{B}$.

Consider the following meta-algorithm. Treat every strategy $S(b)$ as an expert and run a regret minimizing algorithm for learning with expert advice (such as Multiplicative Weights [18] or Follow-The-Perturbed-Leader [17]). The distributions generated by the regret minimizing algorithm are treated as mixing weights for the different strategies, essentially executing each strategy scaled by its current weight. In each round, the meta-algorithm restores the inventory of each strategy to the correct state by additionally buying or selling enough shares so that its inventory is exactly what it would have been had it run the different strategies with their present weights throughout. The specific algorithm is given below.

---
**Algorithm 2** Low regret meta-algorithm
---
1: Run every strategy $S(b)$ in parallel so that at the end of each time period $t$, all trades made by the strategies and the vectors $H_{t+1}$, $C_{t+1}$ and $V_{t+1} \in \mathbb{R}^N$ can be computed.
2: Start a regret-minimizing algorithm $\mathcal{A}$ for learning from expert advice with one expert corresponding to each strategy $S(b)$ for $b \in \mathcal{B}$. Let the distribution over strategies generated by $\mathcal{A}$ at time $t$ be $w_t$.
3: **for** $t = 1, 2, \ldots, T$ **do**
4:     Execute any market orders from the previous period at the current market price $p_t$ so that the inventory now equals $H_t \cdot w_t$. The cash value changes by $-(H_t \cdot (w_t - w_{t-1}))p_t$.
5:     Execute any limit orders from the previous period: a $w_t$ weighted combination of the limit orders of the strategies $S(b)$. The holdings change to $H_{t+1} \cdot w_t$, and the cash value changes by $(C_{t+1} - C_t) \cdot w_t$.
6:     For each strategy $S(b)$ for $b \in \mathcal{B}$, set its payoff in round $t$ to be $V_{t+1}(b) - V_t(b)$ and send these payoffs to $\mathcal{A}$.
7:     Obtain the updated distribution $w_{t+1}$ from $\mathcal{A}$.
8:     Place a market order to buy $H_{t+1} \cdot (w_{t+1} - w_t)$ shares in the next period, and a $w_{t+1}$ weighted combination of the limit orders of the strategies $S(b)$.
9: **end for**

---

We now prove the following bound on the regret of the algorithm based on the regret of the underlying algorithm $\mathcal{A}$. Recall from Lemma 4 the definition of $G := 2\Delta B + \Delta^2$.

**Theorem 2** *Assume that the price sequence has $\Delta$-bounded volatity. The regret of the meta-algorithm is bounded by*

$$\text{Regret}(\mathcal{A}) + \frac{G}{2} \sum_{t=1}^{T} \|w_t - w_{t+1}\|_1.$$

PROOF: The regret bound for $\mathcal{A}$ implies that $\sum_{t=1}^{T}(V_{t+1} - V_t) \cdot w_t \geq \max_{b \in \mathcal{B}} V_T(b) - \text{Regret}(\mathcal{A})$. Lemma 5 shows that the final total value of the meta-algorithm is at least $\sum_{t=1}^{T}(V_{t+1} - V_t) \cdot w_t - \frac{G}{2} \sum_{t=1}^{T} \|w_t - w_{t+1}\|_1$. Thus, the regret of the algorithm is bounded as stated. $\square$

**Lemma 5** *In round $t$, the change in total value of the meta-algorithm equals*

$$(V_{t+1} - V_t) \cdot w_t + H_t \cdot (w_t - w_{t-1})(p_{t-1} - p_t).$$

*Furthermore,* $|H_t \cdot (w_t - w_{t-1})(p_{t-1} - p_t)| \leq \frac{G}{2}\|w_t - w_{t+1}\|_1$.

PROOF:[Sketch] The expression for the change in the total value of the meta-algorithm is a simple calculation using the definitions. The second bound is obtained by noting that all the $H_t(b)$'s are within $B$ of each other by Corollary 1, and thus $|H_t \cdot (w_t - w_{t-1})| \leq B\|w_t - w_{t-1}\|_1$, and $|p_{t-1} - p_t| \leq \Delta$ by the bounded volatility assumption. $\square$

### 4.1 A low regret algorithm based on Mutiplicative Weights

Now we give a low regret algorithm based on the classic Multiplicative Weights (MW) algorithm [18]. Call this algorithm MMMW (Market Making using Multiplicative Weights).

The algorithm takes parameters $\eta_t$, for $t = 1, 2, \ldots, T$. It starts by initializing weights $w_1(b) = 1/N$ for every $b \in \mathcal{B}$. In round $t$, the algorithm updates the weights using the rule

$$w_{t+1}(b) := w_t(b) \exp(\eta_t(V_{t+1}(b) - V_t(b)))/Z_t,$$

for every $b \in \mathcal{B}$, where $Z_t$ is the normalization constant to make $w_{t+1}$ a distribution.

Using Theorem 2, we can give the following bound on the regret of MMMW:

**Theorem 3** *Suppose we set $\eta_t = \frac{1}{2G} \min \left\{ \sqrt{\frac{\log(N)}{t}}, 1 \right\}$, for $t = 1, 2, \ldots, T$. Then MMMW has regret bounded by $13G\sqrt{\log(N)T}$.*

PROOF:[Sketch] By Theorem 2, we need to bound $\|w_{t+1} - w_t\|_1$. The multiplicative update rule, $w_{t+1}(b) = w_t(b) \exp(\eta_t(V_{t+1}(b) - V_t(b)))/Z_t$, and the fact that by Lemma 4, the range of the entries of $V_{t+1} - V_t$ is bounded by $2G$ implies that $\|w_{t+1} - w_t\|_1 \leq 4\eta_t G$. Standard analysis for the regret of the MW algorithm then gives the stated regret bound for MMMW. □

### 4.2 A low regret algorithm based on Follow-The-Perturbed-Leader

Now we give a low regret algorithm based on the Follow-The-Perturbed-Leader (FPL) algorithm [17]. Call this algorithm MMFPL (Market Making using Follow-The-Perturbed-Leader). We actually use a deterministic version of the algorithm which has the same regret bound.

The algorithm requires a parameter $\eta$. For every $b \in \mathcal{B}$, let $p(b)$ be a sample from the exponential distribution with mean $1/\eta$. The distribution $w_t$ is then set to be the distribution of the "perturbed leader", i.e.

$$w_t(b) = \Pr_p[V_t(b) + p(b) \geq V_t(b') + p(b') \ \forall \ b' \in \mathcal{B}].$$

Using Theorem 2, we can give the following bound on the regret of MMFPL:

**Theorem 4** *Choose $\eta = \frac{1}{2G}\sqrt{\frac{\log(N)}{T}}$. Then the regret of MMFPL is bounded by $7G\sqrt{\log(N)T}$.*

PROOF:[Sketch] Again we need to bound $\|w_{t+1} - w_t\|_1$. Kalai and Vempala [17] show that in the randomized FPL algorithm, probability that the leader changes from round $t$ to $t + 1$ is bounded by $2\eta G$. This implies that $\|w_{t+1} - w_t\|_1 \leq 4\eta G$. Standard analysis for the regret of the FPL algorithm then gives the stated regret bound for MMFPL. □

## 5 Experiments

We conducted experiments with stock price data obtained from `http://www.netfonds.no/`. We downloaded data for the following stocks: `MSFT`, `HPQ` and `WMT`. The data consists of trades made throughout a given date in chronological order. We obtained data for these stocks for each of the 5 days in the range May 6-10, 2013. The number of trades ranged from roughly 7,000 to 38,000. The quoted prices are rounded to the nearest cent. Our spread-based strategies operate at the level of a cent: i.e. the windows are specified in terms of cents, and the buy/sell orders are set to 1 share per cent outside the window. The class of spread-based strategies we used in our experiments correspond to the following set of window sizes, quoted in cents: $\mathcal{B} = \{1, 2, 3, 4, 5, 10, 20, 40, 80, 100\}$, so that $N = 10$ and $B = 100$.

We implemented MMMW, MMFPL, simple Follow-The-Leader[2] (FTL), and simple uniform averaging over all strategies. We compared their performance to the best strategy in hindsight. For MMFPL, $w_t$ was approximated by averaging 100 independently drawn initial perturbations.

| Symbol | Date | $T$ | Best | MMMW | MMFPL | FTL | Uniform |
|--------|------|-----|------|------|-------|-----|---------|
| HPQ | 05/06/2013 | 7128 | 668.00 | 370.07 | 433.99 | **638.00** | 301.10 |
| HPQ | 05/07/2013 | 13194 | 558.00 | *620.18* | -41.54 | 19.00 | 100.80 |
| HPQ | 05/08/2013 | 12016 | 186.00 | *340.11* | -568.04 | -242.00 | -719.80 |
| HPQ | 05/09/2013 | 14804 | 1058.00 | **890.99** | 327.05 | 214.00 | 591.40 |
| HPQ | 05/10/2013 | 14005 | 512.00 | *638.53* | -446.42 | -554.00 | 345.60 |
| MSFT | 05/06/2013 | 29481 | 1072.00 | **1062.65** | -1547.01 | -1300.00 | 542.60 |
| MSFT | 05/07/2013 | 34017 | 1260.00 | 1157.38 | 1048.46 | **1247.00** | 63.80 |
| MSFT | 05/08/2013 | 38664 | 2074.00 | 2064.83 | 1669.30 | **2074.00** | 939.10 |
| MSFT | 05/09/2013 | 34386 | 1813.00 | 1802.91 | 1534.68 | **1811.00** | 656.10 |
| MSFT | 05/10/2013 | 27641 | 1236.00 | *1250.27* | 556.08 | 590.00 | 750.90 |
| WMT | 05/06/2013 | 8887 | 929.00 | 694.48 | 760.70 | **785.00** | 235.20 |
| WMT | 05/07/2013 | 11309 | 1333.00 | 579.88 | **995.43** | 918.00 | 535.40 |
| WMT | 05/08/2013 | 12966 | 1372.00 | **1300.47** | 832.80 | 974.00 | 926.40 |
| WMT | 05/09/2013 | 10431 | 2415.00 | **2329.78** | 1882.90 | 1991.00 | 1654.10 |
| WMT | 05/10/2013 | 9567 | 1150.00 | **1001.31** | 7.03 | 209.00 | 707.70 |

Table 1: Final performance of various algorithms in cents. Bolded values indicate best performance. Italicized values indicate runs where the MMMW algorithm beat the best in hindsight.

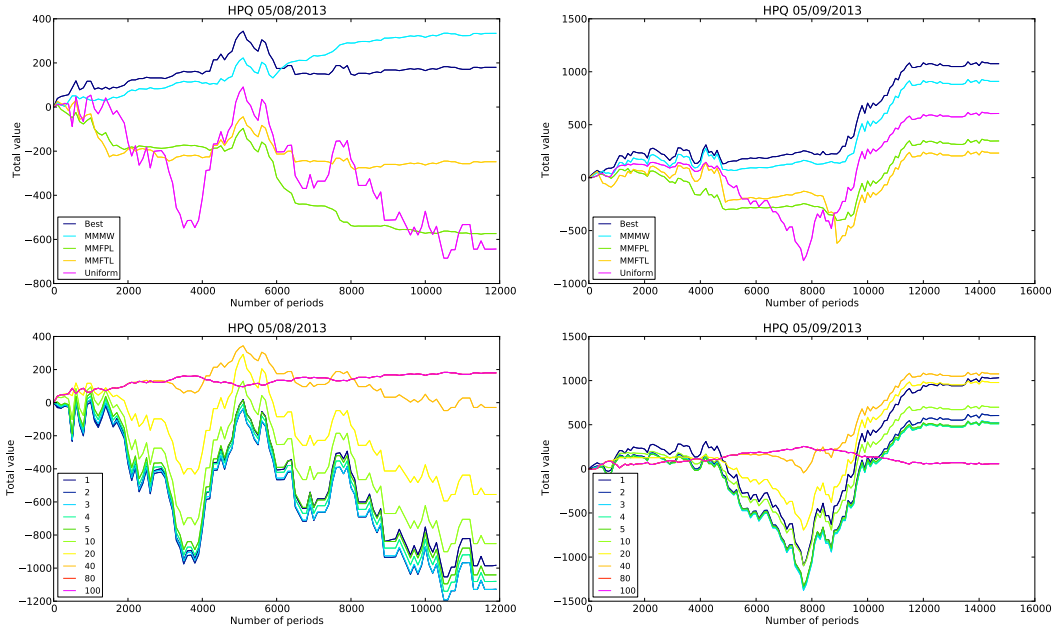

Figure 1: Performance of various algorithms and strategies for HPQ on May 8 and 9, 2013. For clarity, the total value every 100 periods is shown. Top row: On May 8, MMMW outperforms the best strategy, and on May 9 the reverse happens. Bottom row: performance of different strategies. On May 8, $b = 100$ performs best, while on May 9, $b = 40$ performs best.

Experimentally, having slightly larger learning rates seemed to help. For MMMW, we used the specification $\eta_t = \min\left\{\sqrt{\frac{\log(N)}{t}}, \frac{1}{G_t}\right\}$, where $G_t = \max_{\tau \le t, b, b' \in \mathcal{B}} |V_\tau(b) - V_\tau(b')|$, and for MMFPL, we used the specification $\eta = \sqrt{\frac{\log(N)}{T}}$. These specifications ensures that the theory goes through and the regret is bounded by $O(\sqrt{T})$ as before.

Table 5 shows the performance of the algorithms in the 15 runs (3 stocks times 5 days). In all the runs, the MMMW algorithm performed nearly as well as the best strategy, at times even outperforming it. MMFPL didn't perform as well however. As an illustration of how closely MMMW tracks the best performance achievable using the spread-based strategies in the class, in Figure 5 we show the performance of all algorithms for 2 consecutive trading days, May 8 and 9, 2013, for the stock HPQ. We also show the performance of different strategies on these two days - it can be seen that the best strategy differs, thus motivating the need for an adaptive learning algorithm.

## Footnotes

*Work performed while the author was in the CIS Department at the University of Pennsylvania and funded by a Simons Postdoctoral Fellowship

[1]Technically speaking, a brokerage firm won't give the short-seller the cash to spend since this money will be used to backup losses when the short position is closed. But for the purpose of accounting it is perfectly reasonably to record cash in this way, assuming that the strategy ends up holdings at 0.

[2]This algorithm simply chooses the best strategy in each round based on past performance without perturbations.

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
