[Supplementary Material]

# A Omitted proofs

First, we restate and prove Lemma 1.

**Lemma 6** *The value of the portfolio of $S(b)$ at time $T$ can be bounded as*

$$V_{T+1} \geq \frac{\alpha}{\delta} \left( \sum_{t=1}^{T} \frac{b}{2} |a_{t+1} - a_t| - (|a_{T+1} - a_1| + b)^2 \right)$$

PROOF: First let us assume that $\alpha = \delta = 1$ to simplify exposition.

We observe that the holdings $H_t$ is *exactly* the difference $a_1 - a_t$. This is because the number of shares transacted in a period is exactly the distance that the window moves. From this observation it follows that $|V_t - V_{t+1}| \leq |p_t - p_{t+1}| \max(|H_t|, |H_{t+1}|) = |p_t - p_{t+1}| \max(|a_t - a_1|, |a_{t+1} - a_1|)$. In other words, the change in value of the holdings plus cash is no more than the size of the price deviation times whatever was the largest absolute holdings, either at $t$ or $t+1$.

We will use this previous bound to simplify the problem as follows. First let us imagine that on the final round $T$ the price makes a (potentially large) fluctuation to $p_{T+1}$ so that the window returns to its initial point, $a_{T+2} = a_1$. Notice that $p_{T+1}$ can be set so that $|p_T - p_{T+1}| \leq |a_{T+1} - a_1| + b$. The we have:

$$V_{T+1} \geq V_{T+2} - |V_{T+1} - V_{T+2}| \geq V_{T+2} - |p_T - p_{T+1}||a_{T+1} - a_1| \geq V_{T+2} - (|a_{T+1} - a_1| + b)^2.$$

It now suffices to show, with this final price fluctuation returning the window to its starting position, that $V_{T+2} = \sum_{t=1}^{T+1} \frac{b}{2} |a_{t+1} - a_t|$ which will complete the proof. To establish this fact, notice the holdings $H_{T+2}$ finish at $0$ because the window has returned to its initial position, so we can do some clever accouting to determine the cash balance at the final period. We are going to match up every share purchase with a sell, and we can analyze the profit earned from this pair of transactions. We may assume without loss of generality that the price moved by exactly one unit on each round, since we can always take larger price movements and break them up into a sequence of unit trades with no change to the strategy's earnings and holdings. Pick a price $p$, and notice that since the window returned to its initial position $a_1$, then every time point $t$ where $a_t$ moved up to $p$ can be matched up with a time point $t'$ where $a_{t'}$ dropped down to $p - \delta$. When the window moved up to $p$ a share was sold for $p + b$, but when it dropped below $p$ then a share was purchased for $p$. Thus we can record a profit of $b$ for this pair of actions. Put another way, an (amortized) profit of $\frac{b}{2}$ was booked for each unit movement. But the total movement of the window is exactly $\sum_{t=1}^{T+1} |a_{t+1} - a_t|$ as desired. $\square$

Now we restate and prove Lemma 2.

**Lemma 7** *Consider any two strategies $S(b)$ and $S(b')$ with $b' < b$. Let $[a_t', a_t' + b']$ and $[a_t, a_t + b]$ denote the intervals chosen by $S(b)$ and $S(b')$ at time $t$ respectively. Then for all $t$, we have $[a_t', a_t' + b'] \subset [a_t, a_t + b]$.*

PROOF: We prove that the first claim by induction on $t$. For $t = 1$, since $a_1 = a_1' = p_1$, clearly $[p_1, p_1 + b'] \subseteq [p_1, p_1 + b]$. Now assume that for some $t \geq 1$ we have $[a_t', a_t' + b'] \subset [a_t, a_t + b]$. We show that $[a_{t+1}', a_{t+1}' + b'] \subset [a_{t+1}, a_{t+1} + b]$. This follows by easy case analysis on where $p_t$ lies:

1. $a_t' \leq p_t \leq a_t' + b'$. In this case $[a_{t+1}', a_{t+1}' + b'] = [a_t', a_t' + b'] \subset [a_t, a_t + b] = [a_{t+1}, a_{t+1} + b]$.

2. $a_t' + b' < p_t \leq a_t + b$. In this case $[a_{t+1}', a_{t+1}' + b'] = [p_t - b', p_t] \subset [a_t, a_t + b] = [a_{t+1}, a_{t+1} + b]$.

3. $p_t > a_t + b$. In this case $[a_{t+1}', a_{t+1}' + b'] = [p_t - b', p_t] \subset [p_t - b, p_t] = [a_{t+1}, a_{t+1} + b]$.

4. $a_t \leq p_t < a_t'$. Similar reasoning to case 2.

5. $p_t < a_t$. Similar reasoning to case 3.

$\square$

Next, we restate and prove Lemma 3.

**Lemma 8** *For any strategy $S(b)$, its inventory at time $t$, $H_t$, equals $a_1 - a_t$.*

PROOF: We show that $H_t + a_t$ is an invariant. This is easy by case analysis on the position of $p_t$:

1. $p_t \in [a_t, a_t + b]$. In this case neither $H_t$ nor $a_t$ changes so clearly $H_{t+1} + a_{t+1} = H_t + a_t$.
2. $p_t > a_t + b$. In this case, the strategy sells $p_t - (a_t + b)$ shares, so $H_{t+1} = H_t - (p_t - (a_t + b))$, and $a_{t+1} = p_t - b$, so again $H_{t+1} + a_{t+1} = H_t + a_t$.
3. $p_t < a_t$. In this case, the strategy buys $a_t - p_t$ shares, so $H_{t+1} = H_t + a_t - p_t$, and $a_{t+1} = p_t$, so again $H_{t+1} + a_{t+1} = H_t + a_t$.

Thus, $H_t + a_t = H_1 + a_1 = a_1$, and hence $H_t = a_1 - a_t$. $\square$

Next, we restate and prove Lemma 4.

**Lemma 9** *Define $G = 2\Delta B + \Delta^2$. In round $t$, $H = \min_{b \in \mathcal{B}}\{H_t(b)\}$. Then for any strategy $S(b)$, we have*
$$|(V_{t+1}(b) - V_t(b)) - (H(p_t - p_{t-1}))| \leq G.$$
*Thus, for any two window sizes $b$ and $b'$, we have*
$$|(V_{t+1}(b) - V_t(b)) - (V_{t+1}(b') - V_t(b'))| \leq 2G.$$

PROOF: For notational convenience, we will drop the reference to $b$ and simply use $V_t, H_t$ etc. to refer to $V_t(b), H_t(b)$, etc. At any time period $t$, note since $|p_t - p_{t-1}| \leq \Delta$ and $p_t$ lies in the window for $S(b)$, in the next period the strategy buys or sells at most $\Delta$ shares. Thus $|H_{t+1} - H_t| \leq \Delta$. Since the prices paid (or charged) for the $|H_{t+1} - H_t|$ shares bought (or sold) lie between $p_{t-1}$ and $p_t$, so does the average price $p$. The change in total value is

$$
\begin{aligned}
V_{t+1} - V_t &= H_{t+1}p_t - H_t p_{t-1} - (H_{t+1} - H_t)p \\
&= H(p_t - p_{t-1}) + (H_{t+1} - H)(p_t - p) + (H_t - H)(p - p_{t-1}).
\end{aligned}
$$

Now by Corollary 1 we have $|H_t - H| \leq B$. Furthermore, we have $|H_{t+1} - H| \leq |H_{t+1} - H_t| + |H_t - H| \leq \Delta + B$, $|p_t - p| \leq \Delta$, and $|p - p_{t-1}| \leq \Delta$, so

$$|(H_{t+1} - H)(p_t - p) + (H_t - H)(p - p_{t-1})| \leq (\Delta^2 + 2B\Delta) = G.$$

$\square$

Next, we restate and prove Lemma 5.

**Lemma 10** *In round $t$, the change in total value of the meta-algorithm equals*
$$(V_{t+1} - V_t) \cdot w_t + H_t \cdot (w_t - w_{t-1})(p_{t-1} - p_t).$$
*Furthermore,*
$$|H_t \cdot (w_t - w_{t-1})(p_{t-1} - p_t)| \leq \frac{G}{2}\|w_t - w_{t+1}\|_1.$$

PROOF: At the beginning of time period $t$, the meta-algorithm's inventory is $H_t \cdot w_{t-1}$, and at the end, it is $H_{t+1} \cdot w_t$. The cash value changes by $-(H_t \cdot (w_t - w_{t-1}))p_t + (C_{t+1} - C_t) \cdot w_t$ in round $t$. Overall, the change in the total value of the meta-algorithm equals

$$
\begin{aligned}
H_{t+1} \cdot w_t p_t - H_t \cdot w_{t-1}p_{t-1} &- (H_t \cdot (w_t - w_{t-1}))p_t + (C_{t+1} - C_t) \cdot w_t \\
&= (V_{t+1} - V_t) \cdot w_t + H_t \cdot (w_t - w_{t-1})(p_{t-1} - p_t),
\end{aligned}
$$

since $V_t = C_t + H_t p_{t-1}$ and $V_{t+1} = C_{t+1} + H_{t+1} p_t$.

Now, let $H := \min_{b \in \mathcal{B}}\{H_t(b)\}$. By Corollary 1, we have $|H_t(b) - H| \leq B$ for all $b$, so if we define the vector $D$ as $D(b) = H_t(b) - H$, then $\|D\|_\infty \leq B$. Note that $H_t \cdot (w_t - w_{t-1}) = D \cdot (w_t - w_{t-1})$, so

$$|H_t \cdot (w_t - w_{t-1})(p_{t-1} - p_t)| = \|D\|_\infty \|w_t - w_{t-1}\|_1 |p_{t-1} - p_t| \leq B\|w_t - w_{t-1}\|_1 \Delta \leq \frac{G}{2}\|w_t - w_{t-1}\|_1.$$

$\square$

Next, we restate and prove Theorem 3.

**Theorem 5** *Suppose we set $\eta_t = \frac{1}{2G} \min\left\{\sqrt{\frac{\log(N)}{t}}, 1\right\}$, for $t = 1, 2, \ldots, T$. Then MMMW has regret bounded by $13G\sqrt{\log(N)T}$.*

PROOF: By Lemma 4, the range of the entries of $V_{t+1} - V_t$ is bounded by $2G$. Since the choice of $\eta_t$ ensures that $2\eta_t G \leq 1$, we have the following bound on the regret of the MW algorithm:

$$\text{Regret(MW)} \leq \sum_{t=1}^{T} 4\eta_t G^2 + \frac{\log(N)}{\eta_T}.$$

We now bound $\|w_{t+1} - w_t\|_1$. Note that

$$w_{t+1}(b) = w_t(b) \exp(\eta_t(V_{t+1}(b) - V_t(b)))/Z_t.$$

By Lemma 4, we conclude that due to the normalization by $Z_t$, we have

$$\exp(-2\eta_t G) \leq \frac{w_{t+1}(b)}{w_t(b)} \leq \exp(2\eta_t G).$$

Thus, since $\eta_t \leq 1/2G$, we have $|w_{t+1}(b) - w_t(b)| \leq 4\eta_t G w_t(b)$, and so $\|w_{t+1} - w_t\|_1 \leq 4\eta_t G$.

Thus, by Theorem 2, we get the following bound on the regret of MMMW:

$$\text{Regret(MMMW)} \leq \sum_{t=1}^{T} 6\eta_t G^2 + \frac{\log(N)}{\eta_T} \leq 13G\sqrt{\log(N)T},$$

using the choice $\eta_t = \frac{1}{2G} \min\left\{\sqrt{\frac{\log(N)}{t}}, 1\right\}$. $\square$

Finally, we restate and prove Theorem 4.

**Theorem 6** *Choose $\eta = \frac{1}{2G}\sqrt{\frac{\log(N)}{T}}$. Then the regret of MMFPL is bounded by $7G\sqrt{\log(N)T}$.*

PROOF: Fix the initial perturbation $p$. For every round $t$, let the "perturbed leader" $\ell(t)$ be the strategy $S(b)$ such that $V_t(b) + p(b)$ is maximum over all $b \in \mathcal{B}$ (breaking ties arbitrarily).

By Lemma 4, the range of the entries of $V_{t+1} - V_t$ is bounded by $2G$. Kalai and Vempala [17] show that the probability that the leader changes from round $t$ to $t + 1$ is bounded by $2\eta G$, i.e.

$$\Pr_p[\ell(t) \neq \ell(t+1)] \leq 2\eta G.$$

Noting that $w_t = \mathbb{E}_p[e_{\ell(t)}]$, where $e_b$ is the standard basis vector which has 1 in coordinate $b$ and 0 elsehwere, we get that

$$\|w_t - w_{t+1}\|_1 = \|\mathbb{E}_p[e_{\ell(t)} - e_{\ell(t+1)}]\|_1 \leq \mathbb{E}_p[\|e_{\ell(t)} - e_{\ell(t+1)}\|_1] = 2\Pr_p[\ell(t) \neq \ell(t+1)] \leq 4\eta G.$$

Again since the range of the entries of $V_{t+1} - V_t$ is bounded by $2G$, the analysis of Kalai and Vempala [17] shows the following bound on the regret of the FPL algorithm:

$$\text{Regret(FPL)} \leq 4\eta G^2 T + \frac{\log(N)}{\eta}.$$

Thus, by Theorem 2, we get the following bound on the regret of MMFPL:

$$\text{Regret(MMFPL)} \leq 6\eta G^2 T + \frac{\log(N)}{\eta_T} \leq 7G\sqrt{\log(N)T},$$

using the choice $\eta = \frac{1}{2G}\sqrt{\frac{\log(N)}{T}}$. $\square$