[Reviews · NeurIPS 2013]

Submitted by Assigned_Reviewer_8

This paper proposes and analyses (both theoretically and empirically) a class of market making mechanisms that seek to provide liquidity to a market while making a profit through the control of a bid-ask spread. Crucially, this is done without assuming that the prices are due to some stochastic process. The key trick is to make use of existing results for learning with expert advice algorithms where, in the market setting, each expert corresponds to a fixed bid-ask spread strategy. By assuming that market prices are exogenous and of bounded volatility the authors show how guarantees for algorithms such as follow the perturbed leader and multiplicative weights can be adapted to provide guarantees on the regret on the value of the market maker's position relative to the fixed spread experts. Experimental work show the resulting algorithm working in practice on history stock price data.

Despite there being a glaring typo in the title of the paper, the presentation is overall of a high standard. All the key terms are clearly motivated and defined and the proofs are clear and correct. The relevant literature is adequately surveyed and demonstrates that the existing results are novel and significant.


## Other Suggestions

1. Although discretisation is a reasonable way to simplify the analysis, it was not immediately clear how this affects the bounds. Some discussion about how the choice of $\delta$ and $M$ will change the bound in Theorem 2 for a fixed choice $B$ would be enlightening.

2. I would have liked to have seen some discussion of alternative strategies to discretisation. For example, why are results over continuous spaces of experts (e.g., online convex optimisation) applicable here?

3. On line 258 it would be worth explicitly saying that the vectors $H_t$ etc. are in $\mathbb{R}^N$.
Summary: This is a well written paper that clearly demonstrates the potential of applying online "learning with experts" algorithms to designing market mechanisms. The framework introduced here should form the basis for other future work.

Submitted by Assigned_Reviewer_9

The paper presents a nice algorithm for trading a commodity (e.g. stock of certain type) for a market maker (i.e. an entity that must buy from/sell to everybody). The paper nicely explains a (somewhat simplified) mechanics of the stock exchange (limit orders and market orders).

The paper presents a natural set of base strategies. The set of base strategies is parametrized by a parameter "b" -- the spread between buy and sell price. On top of these strategies they authors apply standard online algorithms (Randomized Weighted Majority Algorithm, Follow the Perturbed Leader) for hedging/switching between the base strategies. What makes the application non-trivial is that any strategy (base or not) is stateful -- namely it maintains "cash" and "inventory" (i.e. a number of underlying stocks). Since standard algorithms "for learning with expert advice" are stateless, this makes their application somewhat non-trivial. To overcome this authors prove that certain interesting structural properties of set of base strategies (e.g. the buy-sell price interval of a base strategy with smaller spread is contained in the interval of a strategy with bigger spread). This allows the authors to derive a bound on the regret (i.e. convergence) relative to the best base strategy in hindsight.

The paper is very clearly written. I like it very much and I am happy to recommend it for acceptance.

I have to admit that I don't have time to go through the proofs in detail, but all lemmas seem plausible. Also, I am not familiar with the mathematical finance literature, so I am not sure how interesting this result is within finance context. However, within context of online learning this is a novel, and very interesting result.

1) Please fix the typo ("Marking" -> "Market") in the title of the paper -- it's
embarrassing :)

2) Please be consistent in typesetting authors' names in the references. There are missing dots (e.g. "Robert E Schapire"). Sometimes the first name of a person is present, sometimes there's just the initial ("S. Das" vs "Sanmay Das").

Couple of random ideas for future research:

- It seems that regret of your algorithm depends on the number of switches between the underlying base strategies. There are few recent COLT papers with variants of Randomized Weighted Majority Algorithm (i.e. Hedge/Exponential Weights Algorithm) and Follow the Perturbed Leader that achieve small number of switches between experts. (see e.g. "Regret Minimization for Online Buffering Problems Using the Weighted Majority Algorithm", COLT 2010 and "Prediction by random-walk perturbation", COLT 2013).

- Would it be possibly to get rid of the delta-discretization? The discretization seems to be just an ugly artifact of the model/algorithm. Or to put it differently, what happens in the limit delta --> 0?
Summary: This is a very nice and novel paper about online algorithms for a market maker on a stock exchange (or a commodity market). It contains elegant mathematical model, algorithm and nice theoretical results about the algorithm.

Submitted by Assigned_Reviewer_10

The paper considers a market making problem as described below.
The trader maintains two variables, H_t in R (inventory) and
C_t in R (cash), which are initially set to 0.
For each trial t = 1,...,T, the following happens:
1. Observe a market price p_t in the discrete set PI = {d, 2d, ..., M}.
2. (Market order)
The trader chooses X in R, and then H_{t+1} = H_t + X and C_{t+1} = C_t - p_t X.
3. (Execute Limit Order, skip if t = 1)
The trader executes the limit order, specified by a function L_t: PI to R+,
which is placed in the previous trial.
If p_t > p_{t-1}, then
H_{t+1} -= \sum_{p in PI, p_{t-1} < p <= p_t} L_t(p) and
C_{t+1} += \sum_{p in PI, p_{t-1} < p <= p_t} p L_t(p); and
if p_t < p_{t-1}, then
H_{t+1} += \sum_{p in PI, p_t <= p < p_{t-1}} L_t(p) and
C_{t+1} -= \sum_{p in PI, p_t <= p < p_{t-1}} p L_t(p).
4. (Place Limit Order)
The trader choose a function L_{t+1}: PI to R+ such that L_{t+1}(p_t) = 0.
5. (Portfolio)
The portfolio of the trader becomes V_{t+1} = C_{t+1} + p_t H_{t+1}.

The goal of the trader is make the final portfolio V_{T+1} as large
as possible.

The paper proposes a simple class of strategies for the limit order,
called the spread-based strategies. Each strategy is specified by
a single parameter b in {d, 2d, 3d, ..., Nd}.
Then, the paper proposes an algorithm that combines the spread-based
strategies using a method of low-regret online prediction algorithm
(such as Hedge and FPL) to make its own strategy for the limit
order. The algorithm also employs an appropriate strategy for the
market order and then achieves an O(sqrt(T log N)) regret bound.

The reduction from the market making problem to online prediction
is non-trivial and this work may open a new application area
of online prediction methods.

The cons are that the regret is measured only with respect to
the limit order with a fixed strategy for the market order.
I wonder whether we can also consider a natural strategy class for
the market order and derive an algorithm that is competitive with
the best offline strategies for the market order as well as for
the limit order.
Moreover, the paper does not provide a new method from the
viewpoint of machine learning.

The title of the paper may have a typo.
Summary: This work is a nice application of online prediction.
Perhaps it could be better when presented at a more appropriate
conference like EC.
Author Feedback

Author rebuttal: We thank the reviewers for their careful reading of the paper and their constructive suggestions - especially for catching the typo in the title, which is indeed embarrassing and will be fixed promptly!

Assigned_Reviewer_10:
Market order based strategies: We did not consider competing with strategies that only set market orders. This is perhaps not a bad suggestion, but it didn't match with the main goals of the work, which was to compete with other market making strategies. By definition, a market maker makes both buy and sell offers simultaneously, and we considered strategies within this realm.

Assigned_Reviewer_8:
Discretization: The discretization parameter delta makes sense only in conjunction with the liquidity density parameter alpha. For every unit price change, the strategies trade alpha/delta shares. To compare different discretization parameters delta, it makes sense to also adjust alpha so that alpha/delta is a fixed value so that the number of shares traded per unit price change is the same. The regret bound of Theorem 2 for general alpha and delta is scaled by the same factor, viz. alpha/delta, and so it remains unchanged with changing discretization. We agree this should have further discussion in the final version which we intend to add.

Continuous spaces of experts: This is a good idea, one we considered. Unfortunately, the payoff functions are not convex (or rather, concave) so online convex optimization cannot be applied here.

Assigned_Reviewer_9:
COLT papers on minimizing switches between experts: yes we have considered utilizing such switching strategies. But the results mentioned have the same expected number of switches, viz. O(sqrt{T log(N)}). Thus they don't provide an improvement over what we have in the present paper.